# Ocbil Theory as a Potential Unifying Framework for Investigating Narrow Endemism in Mediterranean Climate Regions

**DOI:** 10.3390/plants12030645

**Published:** 2023-02-01

**Authors:** Stephen D. Hopper

**Affiliations:** Albany Centre, School of Agriculture and Environment, The University of Western Australia, Albany, WA 6330, Australia; steve.hopper@uwa.edu.au

**Keywords:** ancient landscapes, climatically buffered, infertile soils, Southwest Australian Florisitc Region, Greater Cape Floristic Region, speciation, extinction

## Abstract

OCBIL theory addresses the ecology, evolution, and conservation of biodiversity and cultural diversity on old climatically buffered infertile landscapes, which are especially prominent in southwest Australia and the Greater Cape Region of South Africa. Here, as a contribution to general theory on endemism, a few case studies are briefly discussed to ascertain the relevance of hypotheses in OCBIL theory to understanding narrow endemism in Mediterranean climate regions. Two new conservation management hypotheses are also introduced—minimising disturbance of OCBILS and conserving cross-culturally to achieve best outcomes. Case studies of endemics in southwest Australia (e.g., *Eucalyptus caesia*, *Anigozanthos*, Cephalotaceae, Daspypogonaceae) and South Africa (*Moraea*, *Conophytum*) and more limited evidence for the Mediterranean Region conform to OCBIL theory predictions. Narrow endemics, concentrated in OCBILs, have diverse origins that embrace major hypotheses of OCBIL theory such as prolonged persistence and diversification in refugia, limited dispersal, coping with inbreeding in small disjunct population systems (the James Effect), special adaptations to nutrient-deficient soils, and special vulnerabilities (e.g., to soil disturbance and removal). Minimising disturbance to OCBILs is recommended as the primary conservation strategy. OCBIL theory has a potentially significant role to play in advancing understanding of narrow endemism of plants in Mediterranean climate regions and elsewhere.

## 1. Introduction

Exceptional narrow endemism has long been recognised as a feature of Mediterranean climate regions. As of yet, a general theory on endemism has largely remained elusive. Attempts to explain it have usually focused on single-factor hypotheses. Narrow endemism is attributed to soil mosaics, pollination systems, or altitudinal variation for example. Yet, each factor invoked as being the primary cause is easily challenged through identification of exceptions. It has become clear that more than single causes are needed to be invoked to derive satisfactory explanatory hypotheses.

After decades of investigation of patterns of endemism in the Southwest Australian Floristic Region (SWAFR sensu [1,2]) it was realised that three factors in combination appear critical to understanding narrow endemism—landscape age, minimised disturbance regimes (especially climatic buffering due to prolonged oceanic proximity), and infertile soils (especially those low in P). This was published in the progressive development of the OCBIL theory [3,4,5,6], which addresses the ecology, evolution, and conservation of biodiversity and cultural diversity on old climatically buffered infertile landscapes (OCBILs). The opposite of OCBILs are YODFELs [3], or young often disturbed fertile landscapes, usually lowlands adjacent rivers, lakes, or coastlines, but also plains enriched by volcanic eruptions, glaciation and dust storms, and steep mountain slopes. YODFELs are more abundant worldwide than OCBILs, especially in the Northern Hemisphere where most biologists live and work. OCBILs are predominantly though not exclusively found in the Southern Hemisphere and have not enjoyed detailed scientific scrutiny until relatively recently. Consequently, much has to be learnt about the relevance and applicability of OCBIL theory to conserving biodiversity at global scale.

Three regions were selected in the initial OCBIL paper [3] to illustrate aspects of the theory—the SWAFR; the Greater Cape Floristic Region of South Africa [7]; and the Pantepui of equatorial Venezuela, Brazil, and the Guianas [8]. Currently, it is recognised that OCBILs are found in half of the world’s 36 Global Biodiversity Hotspots [5]. The theory has been independently examined and found to be heuristic in a number of new regions, exceptionally so, for example, with campo rupestre altitudinal grassland in eastern South America [9].

The unifying framework of OCBIL theory centred around old, climatically buffered, and infertile landscapes has been challenged, particularly early in the exploration of OCBIL theory [10]. However, subsequent reviews and new evidence have upheld the foundational tenets of OCBIL theory [5,6,11]. Yet, at such an early stage, there are clear biases in OCBIL regions examined and hypotheses tested [11]. More work on the general applicability of OCBIL theory is needed.

As of yet, a review of Ocbil theory as a unifying framework for investigating narrow endemism in Mediterranean climate regions has not been undertaken. Here, I aim to do so briefly, first summarising the framework of hypotheses for OCBIL theory and then highlighting a few case studies from Mediterranean climate regions (MCRs) to stimulate further research on this promising corpus of theory.

## 2. Hypotheses of OCBIL Theory

As of this paper, 12 ecological, evolutionary, and cultural/anthropological hypotheses and another 12 management hypotheses (including two newly proposed herein) provide a predictive and testable body of knowledge within OCBIL theory (Figure 1). Each hypothesis is explored and explained elsewhere [3,4,5,6] with relevant citations, as space does not allow for anything but preliminary mention herein. However, it is important to note that a new contribution in Figure 1 is the addition of two management hypotheses not mentioned by [6].

These two new management hypotheses are minimise human disturbance, and conserve cross-culturally. The additional hypotheses have emerged from ongoing research, especially cross-cultural research on landscape management with First Nations people (e.g., Lullfitz et al. 2021; Merningar Bardok Elder Lynette Knapp, pers. comm.). Our research, and that of many other ethnographic researchers, have found that First Nations revere sacred uplands. This reverence ensures conservation of OCBILs as minimal human disturbance is often practised on them. The prolonged relative stability of OCBIL habitats necessitates this minimising human disturbance for biodiversity conservation. Thus, conserving cross-culturally is promising to offer profound new perspectives in conservation biology [12,13]- that may well halt and possible reverse the ongoing decline of biodiversity. For example, the world view articulated by SWAFR Elder Lynette Knapp [11] clearly divided landscapes into *kaat* (upland hills and mountains) and *beeliar* (freshwater streams and lakes). This fundamental division was recognised culturally in diverse ways, ensuring that human disturbance was concentrated in the lowland *beeliar* (YODFELs) and avoided in the upland *kaat* (OCBILs). Application of this world view has undoubtedly played a significant role in the conservation of narrow endemics and threatened species in the SWAFR.

Increased local endemism is featured in Figure 1 and its causes are highlighted in other OCBIL hypotheses including reduced dispersability, common rarity, accentuated persistance of old lineages, the James Effect, semiarid speciation cradles, reduced hybridisation, and nutritional and other specialisation. This diversity of proposed causes of local endemism is embraced in OCBIL theory.

## 3. Mediterranean Climate Regions and OCBILs

Two of the five Mediterranean Climate Regions (MCRs) have long been recognised as containing old climatically buffered infertile landscapes—the Southwest Australian Floristic Region [1,2] and the Greater Cape Floristic Region of South Africa [7]. These ancient southern hemisphere regions exhibit pronounced geological, topographic, and relative climatic stability [14]. Until recently, the other three MCRs have escaped detailed examination from the perspective of OCBIL theory. However, recent scrutiny has led some authors to suggest that OCBILs may indeed exist in parts of California and Spain [11]. Very preliminary investigations also suggest that the Valdivian rainforest of Chile’s coastal range may also harbour OCBILs (Hopper, unpubl.). Further work is needed to verify or refute these suggestions. For the purposes of this review, I will focus on the SWAFR and GCFR, and briefly on the Mediterranean Floristic Region hereon.

## 4. Narrow Endemism, OCBILs, and the Southwest Australian Floristic Region

Extensive research on rare and threatened species in the SWAFR has now been undertaken for half a century. A noteworthy outcome of early biogeographical survey established that some 64% of well-documented local endemics of conservation interest are confined to upland OCBILs in the region, notably hills of granite, quartzite, and ironstone, as well as upland sandplain (*kwongkan* in the local Noongar dialects) in the broad subdued valleys of the wheatbelt and adjacent Great Western Woodlands region (Figure 2; [6,15]).

Many of the attributes predicted for OCBIL species are evident in SWAFR narrow endemics. Gravity dispersed seed is seen in a high proportion of the flora with no obvious means of attracting animals nor the wind as active dispersal agents [3]. This fundamental feature of OCBIL endemics leads to predictions of common rarity and high levels of population divergence on the insular upland habitats the endemics commonly occupy. This is seen in *Eucalyptus caesia* (Figure 3), an endemic of central wheatbelt granite inselbergs, where even populations as close together as 7 km display exceptional genetic divergence [16,17,18,19]. It is even more pronounced in *Banksia seminuda* subsp. *remanens* on inselbergs of the south coast, displaying much higher divergence over its 40 km geographical range on OCBILs compared with *B. seminuda* subsp. *seminuda* distributed along younger fertile landscapes of river systems over 400 km ([20]; Figure 3). Moreover, these local endemics display accentuated persistence; old clonal individuals of slow woody growth; the James Effect including purging of deleterious genes in small populations and adaptation to mobile bird pollinators; and nutritional specialisations such as mycorrhizal associations, strong lateral root development in cracks in granite, and cluster root for mining low P levels (refs in above reviews).

Molecular phylogenetic studies have proven vital to establishing that accentuated persistence of old endemic lineages is evident in the SWAFR flora, e.g., Cephalotaceae [21] and Dasypogonales/Dasypogonaceae [22].

## 5. Narrow Endemism, OCBILs, and the Greater Cape Floristic Region

The Greater Cape is renowned for its high levels of endemism at approximately 70%. There are many genera that have speciated prolifically, especially among geophytes and the succulent semi-arid dwelling mesembs (Figure 4). Although few authors have specifically investigated GCFR genera from an OCBIL perspective, ample evidence exists where molecular phylogenetics and chromosomal studies have been undertaken. Limited dispersal capabilities are evident, and a bewildering array of highly localised endemics occur in geophyte and/or succulent genera such as *Moraea* (Iridaceae) and *Conophytum*, respectively, from Namaqualand.

Abundant evidence of such genera with predicted attributes of OCBIL species (Figure 1) is seen on careful examination of the published literature [3,5]. In the widespread *Moraea fugax*, for example [23], an exceptional diversity in chromosome number was documented in a dysploid descending series (n = 10, 9, 8, 7, 6, and 5). Goldblatt [23] (p. 149) proposed that this chromosomal diversity ‘probably has promoted population differentiation in the species by restricting hybridisation and consequent recombination in forms that have differentiated cytologically’. In the context of OCBIL theory, this is a spectacular example of the James Effect, conserving heterozygosity in the face of inbreeding due to small disjunct population structures. It also exemplifies predictions of the reduced hybridisation/hybrid speciation hypothesis, due to the accumulation of genetic differences that form reproductive barriers to hybridisation over prolonged periods of isolation on OCBILs.

In the second example illustrated in Figure 4, *Conophytum*, perhaps the most spectacular example of narrow endemism in the GCFR flora, is seen. With more than 108 species and at least 57 subspecies, *Conophytum* has an exceptional 28% of all taxa occurring as single-point endemics [24]. Unlike the dysploidy seen in *Moraea fugax*, *Conophytum* displays exceptional diversity in endopolyploidy. ‘Leaf and flower tissues of *Conophytum* are highly polysomatic and ploidy states of 2C–64C were typically observed across the genus, with some instances of 128C.’ [25]. Essentially, this duplication of whole genomes observed across 46 of the 108 species of *Conophytum* has the same effect as dysploidy does in *Moraea fugax*. It is another example of the James Effect and of the reduced hybridisation hypothesis of OCBIL theory. Scarcely investigated in the GCFR, polyploidy at exceptional infraspecific levels is also seen in *Oxalis obtusa* [26]. Although much more work is needed in the GCFR to fully understand the relevance of OCBIL theory, such investigations and others cited elsewhere [3,4,6] have already confirmed the merits of this body of theory in helping understand narrow endemism.

Molecular phylogeny has enabled penetrating insights on endemism and accentuated persistence of old lineages in GCFR taxa such as in *Protea*, *Moraea* [27], and Brunniaceae [28].

## 6. Narrow Endemism, OCBILs, and the Mediterranean Floristic Region

The extensive literature on evolution of the Mediterranean flora summarised by Thompson [29] and others reveals that an estimated 60% of the 25,000 Mediterranean species are endemics, with 37% recorded as local endemics. Some 20% occur in the Baetic-Rifan Global Biodiversity Hotspot of the western Mediterranean flanking either side of the Gibralter Strait, and 27% are in the Greek mountains (Figure 5). Dispersal limitation is evident in Mediterranean cliff endemics (Thompson 2020). The occurrence in Andulusian Spain of *Drosophyllum lusitanicum*, the monotypic representative of the Mediterranean flora’s only endemic plant family, exemplifies the predicted accentuated persistence of old lineages characteristic of OCBILs (Figure 1 and Figure 5). Indeed, examining and testing the evolution of endemics in the MFR in the context of OCBIL theory is commended as a promising line of general research bearing on the theory of endemism and refugia.

## 7. Conservation Management Predictions of OCBIL Theory

The few examples highlighted above serve to illustrate that OCBIL theory shows considerable promise in unravelling aspects of local endemism across Mediterranean Climate Regions. This brief account is intended simply to highlight that potential, as well as to add two additional biological conservation management hypotheses that have been derived from recent studies mainly in the SWAFR—minimising human disturbance and conserving cross-culturally with First Nations Elders. OCBIL theory has matured as a progressive elaboration of hypotheses that have been published. Given the significant contributions on OCBILs coming out of eastern montane campos rupestre vegetation in South America [9], the present paper suggests that Mediterranean Climate Regions of the world deserve similar attention, particularly in the Mediterranean Region itself, in California, and in Chile.

## Figures and Tables

**Figure 1 plants-12-00645-f001:**
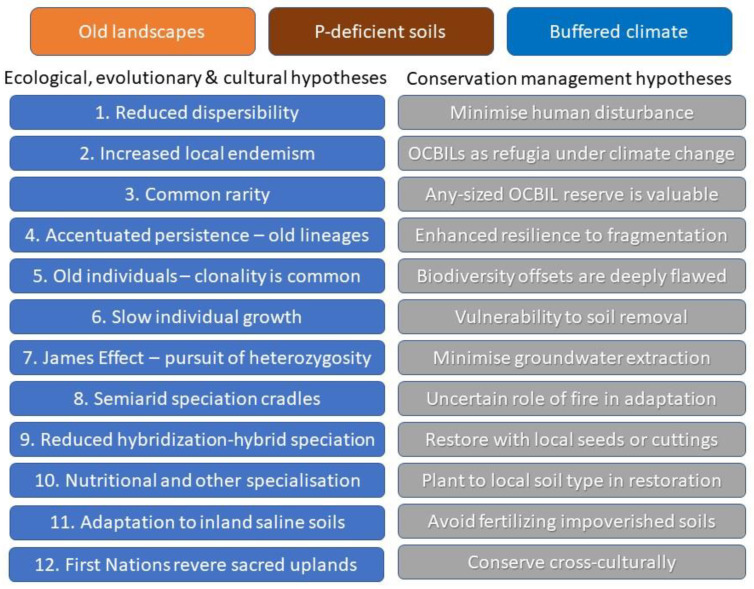
OCBIL theory in overview (modified from [6]), according to the foundational attributes that define OCBILs—old landscapes that are climatically buffered with reduced disturbance levels, and characterised by infertile (P-deficient) soils. There are 12 predictive ecological, evolutionary, and cultural hypotheses and 12 conservation management hypotheses that combined provide a testable foundation for OCBIL theory.

**Figure 2 plants-12-00645-f002:**
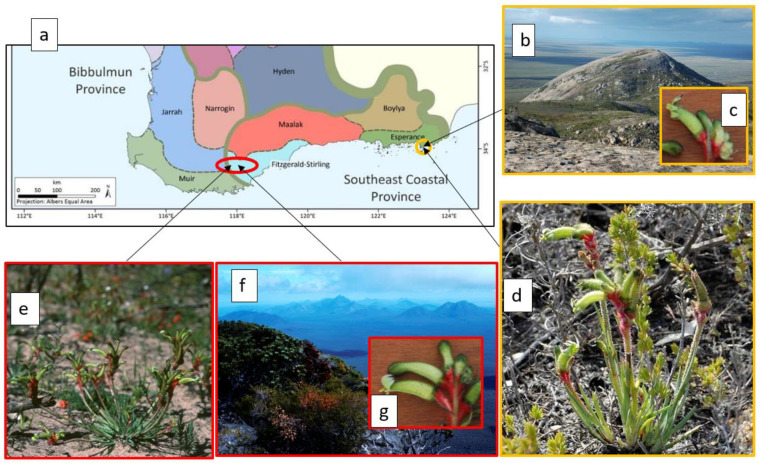
The sister species *Anigozanthos yorlining* Hopper ms and *A*. *gabrielae* Domin: (**a**) distributions in the context of floristic provinces and regions and districts from Gioia and Hopper (2017); (**b**–**d**) *A. yorlining* at granite OCBIL Mt Arid; (**e**–**g**) *A. gabrielae* plains adjacent Stirling Range OCBIL quartzites. Photos S.D. Hopper.

**Figure 3 plants-12-00645-f003:**
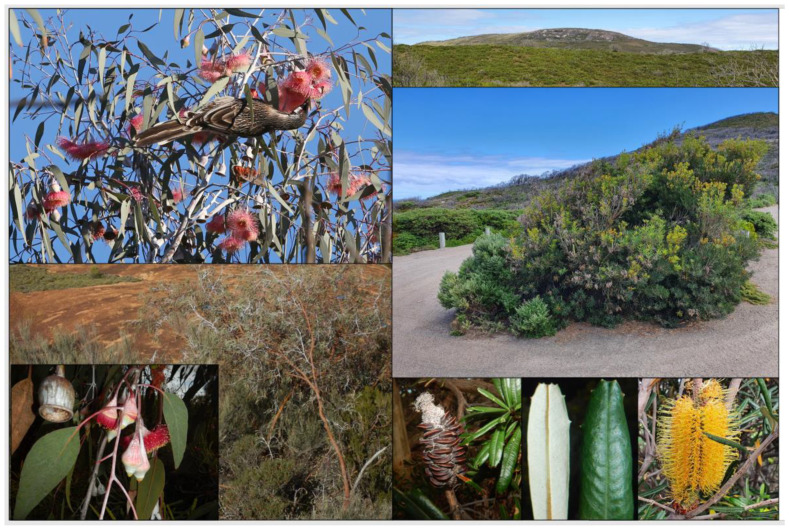
Two examples of narrow endemics from the SWAFR whose biological attributes and granite outcrop habitats occupied meet those predicted by OCBIL theory. Left: *Eucalyptus caesia* (Myrtaceae) and right: *Banksia seminuda* subsp. *remanens* (Proteaceae). Photos S.D. Hopper, Keith Lightbody (top left), and Tim Robins (bottom right).

**Figure 4 plants-12-00645-f004:**
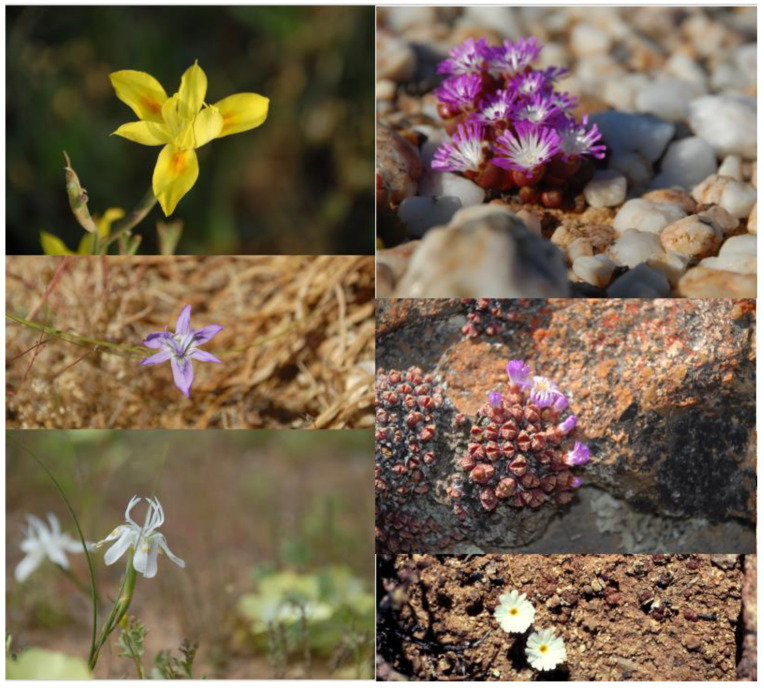
Two examples of genera rich in narrow endemics from the GCFR *Moraea fugax* (Iridaceae, left column; white flower is of *M*. *filicaulis*, regarded earlier as a subspecies of *M*. *fugax*) and *Conophytum* from semiarid succulent karoo in Namaqualand (Aizoaceae).—*C. subfenestratum* (top right), *C. kamiesbergense* (middle right), *C. pellucidum* (bottom right). Photos S.D. Hopper.

**Figure 5 plants-12-00645-f005:**
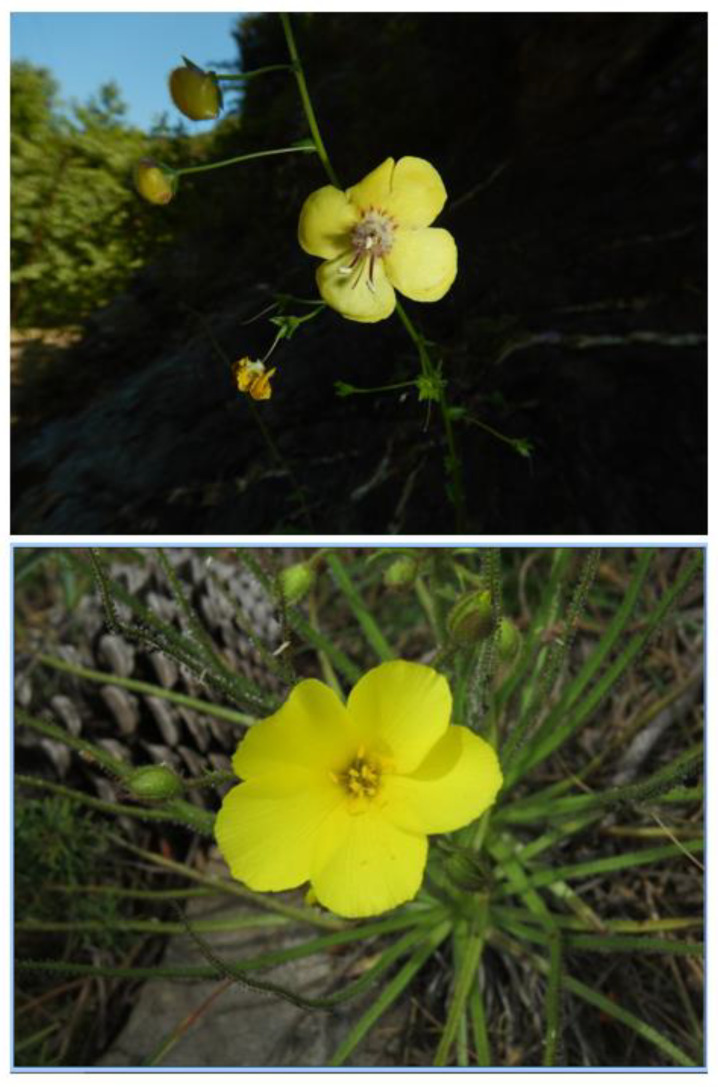
Top: *Verbascum peraffine*, an example of a Mediterranean local endemic confined to mountains and cliff faces on just two Greek islands (southern Evia and adjacent Andros Island to the south; photo S.D. Hopper). Bottom: *Drosophyllum lusitanicum* from southwest Spain, the only plant family endemic to the Mediterranean Floristic Region (photo John Vaclav).

## Data Availability

Original data relevant to this Perspective article are in the references cited.

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
