# Peer review of "Ocbil Theory as a Potential Unifying Framework for Investigating Narrow Endemism in Mediterranean Climate Regions"

_plants, 2023, doi:10.3390/plants12030645_

Round 1

Reviewer 1 Report

The manuscript gives an overview on the OCBIL theory (old climatically-buffered infertile landscapes) according to which a set of environmental characteristics could have enhanced speciation leading to narrow endemics. The author initially proposed this framework for southwest Australia, the Cape Region and the Pantepui region (in the Guiana Shield). Subsequently, he has explored the suitability of other regions for this theory. The manuscript is understood as a presentation and update of OCBIL, which aims at encouraging further studies to test it in other regions, particularly the Mediterranean climate regions. There are some additions concerning ‘conservation management hypotheses’. The volume in which this ms. will be published seems an appropriate forum for encouraging authors interested in Mediterranean climate areas.

The theory is of considerable interest and worth of discussion, scrutiny and further exploration. I was at first puzzled by what was the scope and aims of this manuscript since I read in the abstract “case studies are reviewed”. Later (L. 66) it is said that this will be done “briefly”. I sure missed the clue right above the title where it says that this is a ‘perspective’ paper.  But, overall I think the ms. would benefit from a clearer and earlier statement of what the scope and aims are, considering other recent papers published by the author. Along with this, maybe (on L. 10) it would be clearer to say “a few case studies are discussed” or “commented”

Other points

L.30—I would avoid speaking of falsifiability. A hypothesis on evolution (speciation) across groups (in parallel) fostered by abiotic factors is not a universal and thus is not falsifiable in strict epistemological terms. Besides, you highlight the “diversity of proposed causes” for OCBIL (L.100-101) along with the idea that some of these causes might hold in some cases but not in others and yet this wouldn’t necessarily reject the theory. This situation is incompatible with the principle of falsification.  

L.50—After “SWAFR”, I suppose you need a “,” instead of a “.”

L. 113—“For the purposes of this review…”   See my comments above.

L.114—What does “MFR” stand for? On L. 103 you use “MCR”

L.141—I assume “of” is not right. I suppose you mean “The sister species Anigozanthos yorlining Hopper and A. gabrielae”

L.154—Some references of phylogenetic studies on specific groups providing results that support for, or are relevant to, the OCBIL theory would be most appropriate here.

L.170-172—No indication of which species of Conophytum we are seeing on the left. At least you could say “Conophytum spp”. After “GCFR”, a “:” or equivalent sign is needed.

L.163-167—“In the context of OCBIL theory, this is a spectacular example of the James Effect, conserving heterozygosity in the face of inbreeding due to small disjunct population structures, and of the reduced hybridisation/hybrid speciation hypothesis, due to the accumulation of genetic differences that form reproductive barriers to hybridisation over prolonged periods of isolation on OCBILs.” I think this sentence is a bit confusing. In the previous one you allude to the lack of hybridization between populations (by polyploidy or disploidy), which allow population differentiation, not to the James effect. If I understood correctly, the James effect claims that heterozygosity might be maintained in small populations by selection against homozygotes. I suggest that you remove this effect from the sentence and comment its involvement in Moraea in another one.

L.189-199— Since you are briefly discussing here the suitability of the Mediterranean basin for OCBIL theory and climatic changes over time, or lack thereof, are key to this theory, I think it is appropriate mentioning that climatic instability in the Mediterranean basin is clearly higher than in other Mediterranean climate regions such as the Cape Region and SW Australia (e.g., Cowling et al., 2004; Rundel et al. 2016; etc).

L.193—Typo: “Gibralter”

L.194— Typo: “Andulusian”

L.194-197— Reference(s) for Drosophyllum studies would be appropriate here.

Author Response

The manuscript gives an overview on the OCBIL theory (old climatically-buffered infertile landscapes) according to which a set of environmental characteristics could have enhanced speciation leading to narrow endemics. The author initially proposed this framework for southwest Australia, the Cape Region and the Pantepui region (in the Guiana Shield). Subsequently, he has explored the suitability of other regions for this theory. The manuscript is understood as a presentation and update of OCBIL, which aims at encouraging further studies to test it in other regions, particularly the Mediterranean climate regions. There are some additions concerning ‘conservation management hypotheses’. The volume in which this ms. will be published seems an appropriate forum for encouraging authors interested in Mediterranean climate areas. NOTED WITH THANKS

The theory is of considerable interest and worth of discussion, scrutiny and further exploration. I was at first puzzled by what was the scope and aims of this manuscript since I read in the abstract “case studies are reviewed”. Later (L. 66) it is said that this will be done “briefly”. I sure missed the clue right above the title where it says that this is a ‘perspective’ paper.  But, overall I think the ms. would benefit from a clearer and earlier statement of what the scope and aims are, considering other recent papers published by the author. Along with this, maybe (on L. 10) it would be clearer to say “a few case studies are discussed” or “commented” AGREED AND DONE. ABSTRACT EDITED

Other points

L.30—I would avoid speaking of falsifiability. A hypothesis on evolution (speciation) across groups (in parallel) fostered by abiotic factors is not a universal and thus is not falsifiable in strict epistemological terms. Besides, you highlight the “diversity of proposed causes” for OCBIL (L.100-101) along with the idea that some of these causes might hold in some cases but not in others and yet this wouldn’t necessarily reject the theory. This situation is incompatible with the principle of falsification.  AGREED. REPLACED FALSIFIED WITH CHALLENGED

L.50—After “SWAFR”, I suppose you need a “,” instead of a “.” AGREED. DONE

  1. 113—“For the purposes of this review…”   See my comments above. AGREED. DONE

L.114—What does “MFR” stand for? On L. 103 you use “MCR” CORRECTED

L.141—I assume “of” is not right. I suppose you mean “The sister species Anigozanthos yorlining Hopper and A. gabrielae” CORRECTED

L.154—Some references of phylogenetic studies on specific groups providing results that support for, or are relevant to, the OCBIL theory would be most appropriate here. AGREED. DONE HERE AND FOR THE GCFR

L.170-172—No indication of which species of Conophytum we are seeing on the left. At least you could say “Conophytum spp”. After “GCFR”, a “:” or equivalent sign is needed. CAPTION AMENDED

L.163-167—“In the context of OCBIL theory, this is a spectacular example of the James Effect, NOT CLEAR WHAT CORRECTION IS BEIUNG SUGGESTED. DIVIDED LONG SENTENCE INTO TWO.

Reviewer 2 Report

This brief ms. reviews the theory of OCBILs (old climatically-buffered infertile landscapes) as a unifying framework for investigating narrow endemism in Mediterranean climate regions. The ms. first summarizes the framework of hypotheses for OCBIL theory, and then highlights case studies from three Mediterranean climate regions, the Southwest Australian Floristic Region, the Greater Cape Floristic Region, and the Mediterranean Floristic Region, in an effort to stimulate further research.

While I find nothing especially problematic with this contribution, it does not seem to add much if anything new. Perhaps that is appropriate for a contribution to the ‘Perspective’ category of Plants.

The repeated reference to first nations and ethnic knowledge seems out of step with the rest of the theory which rests on a framework of ecological and evolutionary hypotheses. What exactly are ‘cultural hypotheses’? This aspect requires clarification.

The author seems overly concerned with defending OCBIL theory against detractors.

One factual error should be corrected: Conophytum is not a geophyte (i.e., a perennial plant with an underground food storage organ, such as a bulb, tuber, corm, or rhizome), as stated on l. 156.

Author Response

This brief ms. reviews the theory of OCBILs (old climatically-buffered infertile landscapes) as a unifying framework for investigating narrow endemism in Mediterranean climate regions. The ms. first summarizes the framework of hypotheses for OCBIL theory, and then highlights case studies from three Mediterranean climate regions, the Southwest Australian Floristic Region, the Greater Cape Floristic Region, and the Mediterranean Floristic Region, in an effort to stimulate further research. THANKYOU.

While I find nothing especially problematic with this contribution, it does not seem to add much if anything new. Perhaps that is appropriate for a contribution to the ‘Perspective’ category of Plants. WHAT IS NEW ARE THE TWO EXTRA MANAGEMENT HYPOTHESES AND THE OVERVIEW PROVIDED FOR THE FIVE MCRs

The repeated reference to first nations and ethnic knowledge seems out of step with the rest of the theory which rests on a framework of ecological and evolutionary hypotheses. What exactly are ‘cultural hypotheses’? This aspect requires clarification. A FEW EXTRA WORDS ADDED TO CLARIFY. THE FOCUS ON CULTURAL HYPOTHESES IS IN PART DUE TO THE INTRODUCTION OF TWO NEW CONSERVATION MANAGEMENT HYPOTHESES THAT ARE LARGELY DRAWN FROM CROSS CULTURAL PUBLICATIONS. THIS IS MADE CLEAR IN THE TEXT.

The author seems overly concerned with defending OCBIL theory against detractors. AGREED. I HAVE TONED DOWN AND SIGNIFICANTLY REDUCED SUCH DEFENSIVE WORDS

One factual error should be corrected: Conophytum is not a geophyte (i.e., a perennial plant with an underground food storage organ, such as a bulb, tuber, corm, or rhizome), as stated on l. 156. CORRECTED